# Direct and Base Excision Repair-Mediated Regulation of a GC-Rich *cis*-Element in Response to 5-Formylcytosine and 5-Carboxycytosine

**DOI:** 10.3390/ijms222011025

**Published:** 2021-10-13

**Authors:** Nadine Müller, Eveliina Ponkkonen, Thomas Carell, Andriy Khobta

**Affiliations:** 1Institute of Toxicology, University Medical Center Mainz, 55131 Mainz, Germany; nadgeier@uni-mainz.de; 2Department of Chemistry and Biochemistry, Ludwig-Maximilians University Munich, 81377 Munich, Germany; eveliina.ponkkonen@cup.uni-muenchen.de (E.P.); thomas.carell@cup.uni-muenchen.de (T.C.); 3Institute of Nutritional Sciences, Friedrich Schiller University Jena, 07743 Jena, Germany

**Keywords:** DNA demethylation, 5-formylcytosine, 5-carboxycytosine, thymine DNA glycosylase (TDG), base excision repair (BER), gene regulation, epigenetic marks

## Abstract

Stepwise oxidation of the epigenetic mark 5-methylcytosine and base excision repair (BER) of the resulting 5-formylcytosine (5-fC) and 5-carboxycytosine (5-caC) may provide a mechanism for reactivation of epigenetically silenced genes; however, the functions of 5-fC and 5-caC at defined gene elements are scarcely explored. We analyzed the expression of reporter constructs containing either 2′-deoxy-(5-fC/5-caC) or their BER-resistant 2′-fluorinated analogs, asymmetrically incorporated into CG-dinucleotide of the GC box *cis*-element (5′-TGGGCGGAGC) upstream from the RNA polymerase II core promoter. In the absence of BER, 5-caC caused a strong inhibition of the promoter activity, whereas 5-fC had almost no effect, similar to 5-methylcytosine or 5-hydroxymethylcytosine. BER of 5-caC caused a transient but significant promoter reactivation, succeeded by silencing during the following hours. Both responses strictly required thymine DNA glycosylase (TDG); however, the silencing phase additionally demanded a 5′-endonuclease (likely APE1) activity and was also induced by 5-fC or an apurinic/apyrimidinic site. We propose that 5-caC may act as a repressory mark to prevent premature activation of promoters undergoing the final stages of DNA demethylation, when the symmetric CpG methylation has already been lost. Remarkably, the downstream promoter activation or repression responses are regulated by two separate BER steps, where TDG and APE1 act as potential switches.

## 1. Introduction

Epigenetic regulation of the genome function is crucial for concerted realization of gene expression programs during development and for maintenance of the lineage-specific gene expression patterns in adulthood. During these processes, functions of the whole genome and of the individual genes are dynamically regulated by deposition and removal or maintenance of epigenetic marks, which include specific chromatin components as well as chemically modified DNA bases [1,2]. 5-methylcytosine (5-mC) at CpG dinucleotides is a major DNA modification in vertebrates, and the only inheritable one. It regulates numerous cellular processes, including tissue-specific gene expression, genomic imprinting and X-chromosome inactivation [3]. Depending on the nature of a specific gene regulatory element (and perhaps cell lineage), a methylated cytosine base can function to promote or preclude recruitment of regulatory proteins by two different mechanisms [4]. Symmetric methylation of CpG sites can be read by the methyl-CpG binding protein (MBP) family members [5,6], which mediate transcriptional repression by recruitment of chromatin-modifying enzymes [7,8,9]. In addition, the methyl mark sometimes acts by directly preventing transcription factor binding to their target sites [10,11,12].

Recent progress in the field of epigenetic reprogramming has revealed that 5-mC can undergo enzymatic oxidation by ten-eleven translocation (TET) family dioxygenases to 5-hydroxymethylcytosine (5-hmC) and further to 5-formylcytosine (5-fC) and 5-carboxycytosine (5-caC) [13,14,15]. The latter two modifications are recognized by the TDG DNA N-glycosylase and cleared from DNA by the base excision repair (BER) pathway [16,17,18]. Newly developed whole-genome sequencing techniques revealed enrichment of 5-fC and 5-caC in promoters of transcribed genes and active enhancers, indirectly suggesting their roles in transcriptional activation [19,20,21]. Still, investigation of the dynamics of the oxidation products of 5-mC at specific genomic loci and its significance for the regulation of transcription is hampered by low abundances of these modifications in the genome and a troublesome discrimination between different cytosine modifications by sequencing techniques [22,23]. It remains a matter of controversy whether 5-hmC should be regarded as a functionally autonomous epigenetic mark, similar to 5-mC, or barely a demethylation intermediate which does not have a dedicated function [22,23,24]. Potential biological functions of the downstream products of the TET-mediated oxidation pathway, 5-fC and 5-caC, are even harder to characterize, as these modifications are much scarcer in genomic DNA [23].

To directly address functional impacts of 5-hmC and its derivatives on the regulation of gene expression, these modifications can be site-specifically incorporated into a suitable reporter vector. We previously used such an approach to investigate the effects of the 5-mC oxidation products in a cyclic adenosine monophosphate response element (CRE). There, 5-mC, 5-hmC, 5-fC and 5-caC all negatively affected the promoter activity by inhibition of the cognate transcription factor CREB binding [25], thus corroborating the mechanism earlier described for 5-mC in the central CpG dinucleotide of the CRE sequence [10]. In addition, 5-fC and 5-caC initiated an indirect silencing mechanism attributed to the base excision and DNA strand cleavage [25]. Thereby, the outcomes of 5-fC and 5-caC in CRE largely recapitulated responses to common types of DNA damage processed by the BER [26,27,28]. It should be kept in mind, however, that binding of regulatory proteins to their cognate target elements in DNA, and consequently the outcomes of these interactions, can be highly context-specific [11,12,29]. Here, we systematically investigated the effects of synthetic 5-mC, 5-hmC, 5-fC and 5-caC incorporated into a GC-rich regulatory element (GC box), on the promoter activity. To model potential DNA demethylation stages, we placed defined cytosines modifications asymmetrically in either strand of the GC box, opposite to the non-methylated CpG dinucleotide. This led to identification of distinct functional states of the promoter conferred by defined steps of the DNA demethylation pathway.

## 2. Results

### 2.1. Gene Repression in HeLa Cells Induced by 5-fC and 5-caC in the GC Box CpG Dinucleotide

To investigate the impact of 5-mC oxidation products in a GC-rich promoter on the gene expression, we generated enhanced green fluorescence protein (EGFP) reporter constructs containing single 5-mC, 5-hmC, 5-fC or 5-caC residues at the CpG site of an artificial promoter containing the 5′-TGGGCGGAGC-3′ GC box sequence as the only *cis*-regulatory element (Figure 1a). In the pGCbox-W and pGCbox-C plasmid vectors, this GC box enhances the basal gene expression of the downstream *EGFP* gene by the factor of ≥2, compared to a scrambled DNA sequence [30]. Synthetic oligonucleotides containing all types of cytosine modifications were incorporated into the pGCbox-W vector, with efficiencies closely approaching 100% (Figure 1b,c). Moreover, the fractions of correctly ligated vector DNA were >92% for all modifications, which warranted that DNA topology or misalignment of the inserted synthetic oligonucleotides would not affect subsequent measurement of the promoter activity in cells.

Time course analyses of pGCbox-W constructs containing different cytosine modifications at the CpG site in the purine-rich strand showed that 5-mC and 5-hmC did not appreciably influence the *EGFP* gene expression levels in transfected HeLa cells (Figure 1d). In contrast, constructs containing 5-fC and 5-caC showed considerably decreased EGFP expression levels. These effects were present already 6 h after transfection (the earliest time point when the expression could be reproducibly measured) and grew stronger in the course of time. At the initial time point, the decrease of the gene expression was more significant for 5-caC than for 5-fC, whereas subsequent dynamics was somewhat faster for 5-fC, leading to a stronger impairment of the gene expression at the later time points. Thus, relative to the reference construct containing unmodified cytosine, the expression levels decreased from 83.4 ± 6.5% to 21.0 ± 3.7% for 5-fC between 6 and 48 h post-transfection. For 5-caC, the promoter activity decreased from 79.4 ± 6.5% to 32.0 ± 16.7% in the same experiment series.

Also in the pyrimidine-rich DNA strand, the incorporation rates of all cytosine modifications were very close to 100%, with >88% synthetic strands fully and correctly ligated (Figure 2). As in the opposite strand, neither 5-mC nor 5-hmC caused considerable changes of the gene expression levels over the entire time interval, whereas 5-fC and 5-caC caused the gene expression to decline strongly and in a time-dependent manner. Once again, the inhibition of gene expression was initially stronger in case of 5-caC, whereas the 5-fC construct demonstrated a faster decline of the gene expression in the course of time. The 6- and 48-h expression values decreased from 87.2 ± 8.1% to 20.3 ± 3.5% for 5-fC and from 82.0 ± 9.8% to 29.3 ± 6.3% for 5-caC. Combined, the results show that 5-hmC and 5-mC in either strand of the GC box did not tangibly affect the promoter activity over a period of at least 48 h. In contrast, 5-fC and 5-caC clearly perturbed the gene expression in a time-dependent fashion.

### 2.2. BER-Resistant 5-caC and, to Some Extent, 5-fC Directly Diminish the GC Box Activity

The observed dynamic changes of the expression of constructs carrying 5-fC and 5-caC, in contrast to 5-mC or 5-hmC, could likely be attributed to removal of these modifications from DNA. Therefore, to measure direct impacts of 5-fC and 5-caC on the GC box activity, it was necessary to eliminate their repair. Activities of DNA N-glycosylases towards their substrates, including 5-fC and 5-caC, can be efficiently inhibited by deoxyribose fluorination at the C2′ position [25,31]. This motivated us to generate GC box constructs carrying the 2′-(*R*)-fluorinated derivatives of 5-fC and 5-caC. Both modifications were incorporated into the purine-rich DNA strand as efficiently as the respective 2′-deoxy compounds (Figure 3a). Quantification of the EGFP expression levels in transfected HeLa cells showed that 2′-(*R*)-fluorination entirely reversed the repressory effect of 5-fC, as the difference between the 6- and 24-h points disappeared. The expression levels were 81.2 ± 9.8% (at 6 h) and 84.6 ± 8.4% (at 24 h post-transfection), relative to the construct containing cytosine. The respective values for 2′-deoxy 5-fC were 88.0 ± 5.7% and 31.0 ± 4.1% (Figure 3b). The repression of the promoter activity caused by 5-fC between 6 and 24 h post-transfection was statistically highly significant (*p* = 8.6 × 10^−6^, Student’s two tailed heteroscedastic t-test), whereas the effect of 2′-(*R*)-fluorinated 5-fC was not (*p* = 0.62). In the case of 5-caC, the differences between the 6- and 24-h time points were, again, highly significant for deoxynucleotide (*p* = 1.6 × 10^−4^) but not for the 2′-fluorinated analog (*p* = 0.38). We thereby conclude that dynamic changes of the promoter activity induced by 5-fC and 5-caC are the consequence of excision of the modified bases by a DNA N-glycosylase mechanism.

As 2′-(*R*)-fluorinated 5-fC and 5-caC in the GC box constructs showed stable levels of the reporter gene expression, it was now possible to quantify their direct effects on the promoter activity, based on the EGFP expression levels in transfected HeLa cells. We used 24-h values to estimate these effects, since fluorescent cell counts as well as average fluorescence intensity per cell were higher at this time point (Figure 3c). Reduction of the gene expression by 2′-(*R*)-fluorinated 5-fC was minimal yet statistically significant (*p* = 0.035). The effect of 2′-(*R*)-fluorinated 5-caC was much stronger (36.2 ± 9.6% residual gene expression relative to the cytosine control) and statistically highly significant (*p* = 9.3 × 10^−4^). Considering that GC box enhances the gene expression by a factor of 2 to 2.5 over the basal expression level [30], the results indicate that the presence of a single 5-caC in the purine-rich strand abolished the activation attributable to GC box completely or almost completely.

We also analyzed the effects of the 2′-(*R*)-fluorinated 5-fC and 5-caC analogs in the pyrimidine-rich GC box strand (Figure 4). As in the purine-rich strand, 2′-fluorination abolished time-dependent repression by both cytosine modifications, as judged by disappeared differences between the 6- and 24-h expression values. The respective *p*-values were 8.1 × 10^−5^ (5-fC), 0.19 (2′-fluoro 5-fC), 9.1 × 10^−4^ (5-caC) and 0.30 (2′-fluoro 5-caC). The effects of BER-resistant modifications differed slightly between the strands. Thus, 2′-(*R*)-fluorinated 5-fC in the pyrimidine-rich strand did not at all inhibit the GC box activity, showing the 24-h relative expression level of 102.1 ± 4.1% (*p* = 0.38). The inhibitory effect of 2′-(*R*)-fluorinated 5-caC remained highly significant in the pyrimidine-rich strand, with relative expression of 54.7 ± 2.3% (*p* = 3.5 × 10^−5^).

In summary, apart from subtle quantitative differences between the purine- and pyrimidine-rich strands of the GC box, the results indicate that processing of 5-fC and 5-caC by BER plays a key role in the regulation of the gene expression. Over an extended time interval (24 h), in HeLa cells, the role of BER is manifested by dynamic repression of the promoter activity, in contrast to steady expression levels observed in the presence of the repair-resistant analogs. The potent negative effect of 2′-(*R*)-fluorinated 5-caC on the promoter activity strongly suggests that 5-caC functions as a negative regulatory mark in the GC box. The function of 5-fC seems to be different, as this modification has little or no effect on the GC box activity, depending on the DNA strand.

### 2.3. BER of 5-caC in the GC Box Induces a Transient Promoter Activation

The evidence implicating the base excision in the transcriptional repression overall corroborated the idea that the time-dependent decline of the gene expression observed earlier (Figure 1 and Figure 2) was caused by gradual BER-mediated clearance of 5-fC and 5-caC from DNA. However, even though the cumulative outcomes of the excision of 5-fC and 5-caC were manifested after 24 h as repression of the reporter gene, we noticed that the initial effects of 5-caC were the opposite (Figure 3b and Figure 4b). In particular, the 6-h EGFP expression values showed an inverse relationship between the activities of the GC box constructs containing BER-resistant versus BER-sensitive modifications, wherein the expression was significantly enhanced by 2′-deoxy 5-caC compared to its 2′-fluorinated counterpart in both the purine-rich (*p* = 1.2 × 10^−3^) and the pyrimidine-rich DNA strands (*p* = 7.2 × 10^−4^). These results indicate that removal of 5-caC reactivated the GC box promoter at least transiently, whereas subsequent gradual decrease of the EGFP expression (reported in Figure 1 and Figure 2) suggests that activation was followed by transcriptional silencing within the next few hours. Thereby, the response to 5-caC in GC box is different from reported previously in CRE, where TDG-mediated silencing occurred without a preceding activation phase [25].

If the pulse of promoter activation during the first hours is specific to 5-caC, it would provide an explanation to the ostensibly slower silencing kinetics in comparison to the 5-fC constructs (Figure 1 and Figure 2). Indeed, the 5-fC constructs did not display a similar early activation by the BER mechanism. Rather on the contrary, BER-sensitive 2′-deoxy 5-fC in the pyrimidine-rich strand slightly but significantly decreased the expression with respect to its BER-resistant counterpart already after 6 h (*p* = 6.4 × 10^−3^), while the respective values for the purine rich strand did not differ significantly (*p* = 0.28).

### 2.4. The Dynamics of Transcriptional Regulation by 5-fC and 5-caC Is Entirely TDG-Dependent

We next aimed at determining whether the effects described for the 2′-(*R*)-fluorinated analogs of 5-fC and 5-caC hold true also for the modifications in the context of the 2′-deoxyribose backbone. To eliminate BER of 5-fC and 5-caC in HeLa cells, we targeted the *TDG* locus by CRISPR-Cas9 (Figure 5a). This resulted in deletions encompassing the exons 2 to 5 (Appendix A) and conferred complete elimination of the TDG protein expression in eight of the selected clones (Figure 5b).

We next performed time-course expression analyses of GC box constructs containing the whole spectrum of cytosine modifications in in a TDG knockout cell line derived from clone F3. The results showed that dynamic EGFP expression changes characteristic for the parental HeLa cells were completely abrogated by TDG knockout (Figure 5c). In the absence of TDG, 5-caC steadily inhibited promoter activity, in the same manner as its 2′-(*R*)-fluorinated analog (Figure 5c and Appendix A). As this result also closely recapitulated the effect of the 2′-(*R*)-fluorinated analog in the parental HeLa cell line (Figure 3 and Figure 4), we conclude that 2′-fluorination does not influence biological properties of 5-fC and 5-caC beyond stabilization of the N-glycosidic bond. In the case of 5-fC, a rather mild inhibition of transcriptional activation was observable only when the modification was present in the purine-rich strand. This outcome was, again, consistent with results obtained for the 2′-(*R*)-fluorinated analog (Figure 3 and Figure 4; Appendix A).

In an independent TDG knockout clone C11 (not shown), responses to 5-fC and 5-caC were the same as in F3. In contrast, responses in the isogenic NTH1 knockout cell line displayed the same pattern as in the parental HeLa cell line (Figure 5c). We thereby conclude that the phenotype displayed by the TDG knockout clones was specific to the targeted locus. Finally, the expression of constructs carrying 5-mC or 5-hmC essentially did not differ between the cell lines, as expected based on the universally stable character of these modifications (Figure 5c). In summary, the results in TDG knockout cells corroborate the conclusion that unrepaired 5-caC prevents or counteracts the GC box activation in the absence of repair. Besides, the steady character of the effect of 5-caC on the gene expression in the absence of TDG strongly suggests that TDG-dependent BER is by far the most efficient pathway for 5-caC removal in the chosen cell model.

Of note, the results confirmed our previous conclusion that BER is accountable not only for long-term gene silencing induced by 5-fC/5-caC but also for transient activation by 5-caC at the beginning of the time course. Moreover, by comparison of expression levels of the same constructs between cell lines with different TDG states, we now could attribute the activation specifically to TDG, as the 6-h expression values were significantly higher in the parental HeLa cell line than in the derived TDG knockouts (Figure 5c and data not shown). For 5-caC in the purine-rich GC box strand, relative expression level was 57.0 ± 5.8% in the TDG knockout clone F3 versus 84.3 ± 5.6% in HeLa (*p* = 4.2 × 10^−3^). In the pyrimidine-rich strand, it was 62.6 ± 9.4% versus 84.7 ± 7.2% in HeLa (*p* = 3.2 × 10^−2^). The respective expression levels in clone C11 also lied significantly below the HeLa values: 66.9 ± 5.3% (*p* = 1.7 × 10^−2^) for 5-caC in the purine-rich strand and 65.2 ± 5.9% (*p* = 2.2 × 10^−2^) in the pyrimidine-rich strand.

### 2.5. Regulation of the GC Box Promoter by Acytosinic Sites and the Effect of Strand Cleavage

TDG is a monofunctional DNA-N glycosylase that leaves deoxyribose as a reaction product in the DNA strand [17]. Intriguingly, analogous apurinic/apyrimidinic (AP) lesions were implicated in the regulation of GC-rich DNA *cis*-elements previously [30,32]. Therefore, we constructed reporters carrying acytosinic lesions in either of the GC box strands. This was achieved by using oligonucleotides, where 2′-deoxy cytosines at the CpG sites were replaced by two types of tetrahydrofuran AP lesions (Appendix A). Being a close structural analog of the natural AP lesion, tetrahydrofuran (F) is efficiently excised by APE1; however, a combination of F with a phosphorothioate linkage on the 5′ side (SF) yields an APE1-resistant lesion [26,33].

The expression levels of GC box constructs containing APE1-resistant acytosinic sites (SF) were slightly decreased with respect to the counterparts containing unmodified C (Figure 6). In both HeLa and TDG knockout cells, they were in the range of 87–90% at the 6-h time point, regardless of the strand, indicating that AP lesion has a mild inhibitory effect on the promoter activity. The respective values for F lesions were somewhat lower (78–83%), which is probably attributable to their incision by APE1. The difference between the F and SF expression levels grew highly significant by the 24-h points: whilst expression of F constructs intensely declined, the expression levels of both SF constructs decreased only slightly. Consequently, the potent gene silencing response to F is attributable to the phosphodiester bond cleavage (Figure 6).

Endonucleolytic strand cleavage seems to be a rather common inducer of transcriptional repression, as previously reported for AP lesions in other contexts [28,30,32]. In contrast, endonuclease-resistant aguaninic lesions can occasionally cause promoter activation, presumably via a non-processive APE1 binding [30,32]. Based on present results, we deduce that acytosinic lesions at the GC box CpG dinucleotide do not cause a full promoter activation. Nonetheless, they enable substantially higher expression levels than observed in the presence of the unrepairable 5-caC analog (Figure 3 and Figure 4).

## 3. Discussion

Both active and passive DNA demethylation mechanisms must necessarily involve a step when only one cytosine in a given double-stranded CpG dinucleotide is modified (methylated, hydroxymethylated, formylated or carboxylated) at the C5 position. Recognition of CpG sites by methyl-CpG-binding domain (MBD) proteins, including DNMT1, is strongly inhibited in the presence of 5-hmC [34,35,36]. This causes impaired methylation maintenance and leads to hemi-hydroxymethylated sites upon replication [37]. An even more pronounced DNMT1 inhibition was reported by 5-fC and 5-caC, implying that replication of DNA containing these modifications would generate hemi-formylated and hemi-carboxylated CpG sites [38]. Hence, we propose that the impacts of defined cytosine modifications on the GC box activity that were reported here can be extrapolated to model functional outcomes of a range of potential DNA demethylation intermediates. Our results suggested that GC box containing a hemi-methylated CpG dinucleotide (with 5-methyl group present in either DNA strand) is as active as in the absence of any modification (Figure 1, Figure 2 and Figure 5). This is in agreement with biochemical evidence that methylation does not inhibit transcription factor binding to various GC box consensus motifs [39,40]. We further found that GC box remained fully active in the presence of single 5-hmC or 5-fC in the pyrimidine-rich strand, whereas the modifications in the pyrimidine-rich strand seemed to cause only a very minor decrease of the activity (Figure 1, Figure 2, Figure 3, Figure 4 and Figure 5). In contrast, 5-caC appears to be the only cytosine modification within the TET pathway, which causes a strong direct impairment of the GC box activity (Figure 3, Figure 4 and Figure 5). Our results thus suggest that the nature of the modification present at asymmetrically modified CpG dinucleotides could be of a critical functional importance.

The inhibitory effect of 5-caC on the GC box can be relieved by a TDG-dependent mechanism. Thus, our results show that expression of constructs carrying 5-caC is reactivated early upon their delivery to cells, but only if TDG is available (Figure 5) and base excision is unhindered (Figure 3 and Figure 4). In HeLa cells, a short period of promoter activation was followed by the onset of a repressed state, which was TDG-dependent as well. However, this response should be regarded as separate from previous activation, since it required, besides base excision, the presence of a labile phosphodiester linkage 5′ to the target nucleotide (Figure 6). A strand cleavage reaction at the critical position is most likely catalyzed by APE1, which is by far the most important AP endonuclease in human cells, whereas the downstream gene silencing mechanism remains to be elucidated. Similar silencing responses were seen with various BER substrates previously and were attributed to adoption of a repressive chromatin structure after the completion of BER [28,41]. It is intriguing to speculate that TDG and APE1 may be subjected to regulation in particular cell lineages and genomic contexts or, perhaps, by endogenous or exogenous signals. This would allow diversification of transcriptional responses and enable plasticity of the epigenetic states in cells undergoing TET- and TDG-dependent DNA demethylation.

To understand the effects of single intermediates arising during the stepwise DNA demethylation pathway on the promoter activity, the expression levels should be related to the amount of a given cytosine modification present. For DNA modifications undergoing repair in cells, quantification of residual modifications specifically in transcription-competent DNA poses a serious technical challenge, because significant fractions of vector DNA distribute to non-nuclear compartments or undergo dynamic changes of expression due to chromatinization [41]. Nevertheless, under the assumption that the effects of BER-resistant 2′-fluorinated synthetic analogs of 5-fC and 5-caC remain steady, it was possible to derive conclusions about direct effects of these modifications on the GC box activity. Indeed, the expression levels in the presence of these modifications, relative to cytosine controls, were found constant over time (Figure 3 and Figure 4). The same holds for natural (2′-deoxy) modifications, when delivered to repair-deficient host cells (Figure 5). Similarly, the effect of AP intermediate could be inferred based on the results obtained with APE1-resistant SF lesion, which also displayed steady expression levels (Figure 6).

## 4. Conclusions

In summary, our results revealed several levels of GC box regulation by cytosine modifications generated within the active DNA demethylation pathway. GC box upstream from RNA polymerase II core promoter retains its full activity when the CpG dinucleotide is hemi-methylated. Oxidation of 5-mC to 5-hmC and 5-fC is well tolerated; however, the ultimate oxidation product 5-caC behaves as a stable repressory mark in the absence of TDG or if protected from the N-glycosylase activity. TDG restores the GC box activity almost completely. It can be assumed that promoter reactivation occurs as soon as an acytosinic lesion is generated. In support of this notion, modeling of the post-excision step, with the help of AP site analogs, led to very similar effects on the promoter activity (Figure 6). Reactivation by the excision of 5-caC lasted only for a few hours in HeLa cells before getting overturned by a concurrent silencing response, which was elicited by strand cleavage at the AP lesion. Although our data provide no evidence of a more permanent promoter activation by the TDG pathway, such a scenario may take place in another cell type or promoter context. For instance, AP lesion can be protected from endonucleolytic processing by a non-canonical DNA structure [42,43,44,45] or shielded by a specific binding protein [46,47]. Alternatively, signaling downstream from the single strand break generation could be modulated towards a different functional state of the promoter.

## 5. Materials and Methods

### 5.1. Synthetic Oligonucleotides Carrying Cytosine Modifications

Deoxyribo-oligonucleotides containing the specified cytosine modifications were 5′- CATTGCATGGG[C*]GGAGCG and 5′-CATTGCGCTC[C*]GCCCACG (where C* is C, 5-mC, 5-hmC, 5-fC or 5-caC). DNA CE-phosphoramidites Bz-dA, Bz-dC, iBu-dG, dT and Bz-mdC were obtained from Glen Research (Sterling, VA, USA) or Link Technologies (Bellshill, Scotland, UK). Syntheses of the 5-hmC, 5-fC and 5-caC phosphoramidites [48], along with the 2′-(*R*)-fluorinated derivatives of 5-fC and 5-caC [31], were performed as described previously. The solid-phase synthesis, HPLC-purification and MALDI/MS quality-control procedures of the 18-mer deoxyribo-oligonucleotides were performed by using the standard protocols described previously [31]. Synthetic apurinic/apyrimidinic (AP) lesions were tetrahydrofuran with either the phosphodiester (F) or the APE1-resistant phosphorothioate 5′-linkage (SF). Oligonucleotides 5′-CATTGCATGGG[AP]GGAGCG and 5′-CATTGCGCTC[AP]GCCCACG (where AP is F or SF) were purchased from BioSpring GmbH (Frankfurt am Main, Germany).

### 5.2. Generation of Reporter Constructs Containing Cytosine Modifications in the GC Box CpG Dinucleotide

Vectors pGCbox-W and pGCbox-C, allowing substitution of the selected GC box strand with synthetic oligonucleotides, were described previously [30]. Both vectors contain a common GC box motif 5′-TGGGCGGAGC as the only *cis*-regulatory element upstream from the RNA polymerase II transcription initiation site and sustain equivalent levels of the reporter *EGFP* gene expression. Defined modifications targeting cytosines of the CpG dinucleotide were introduced into the purine-rich GC box strand of pGCbox-W or into the pyrimidine-rich strand of pGCbox-C, using the available sites by the Nb.BsrDI nicking endonuclease. The procedure was described in detail previously for introducing 8-oxo-7,8-dihydro-2′-deoxyguanosine (8-oxoG) into the same vectors [30]. Plasmid DNA was cut at two tandem sites by the Nb.BsrDI nicking endonuclease, and the excised native DNA strand fragments were substituted for synthetic oligonucleotides containing the modifications of choice, as described previously for a different promoter context [25]. The efficient incorporation of synthetic DNA strands was verified by inhibition of ligation in the absence of T4 polynucleotide kinase [49]. Percentages of covalently closed DNA in the vector preparations were determined by agarose gel electrophoresis in the presence of 0.5 mg/L ethidium bromide, followed by band quantification, using a GelDoc™ EZ imager and the ImageLab™ software (Bio-Rad Laboratories, GmbH, Munich, Germany), as described previously [28]. The presence of AP lesions was verified by excision analysis, using endonuclease IV (NEB GmbH, Frankfurt am Main, Germany). Constructs were incubated with endonuclease IV (8 U/200 ng plasmid DNA) 1 h at 37 °C in 15 µL buffer composed of 10 mM HEPES (pH 7.5), 200 mM NaCl, 1 mM ethylenediaminetetraacetic acid and 0.1 mg/mL nuclease free bovine serum albumin (NEB). The enzyme was heat-inactivated for 20 min, at 85 °C.

### 5.3. Quantitative Analyses of EGFP Expression in Transfected Cells

HeLa cells used in experiments were clones descending from HELA cervical car-cinoma cell line (German Collection of Microorganisms and Cell Cultures No. ACC 57). The derived TDG and NTH1 knockout cell lines were generated in our lab. Cells exponentially growing in 6-well plates were transfected with mixtures containing 400 ng GC box reporter vector (pGCbox-W or pGCbox-C) and 400 ng tracer pDsRed-Monomer-N1 vector (Clontech, Saint-Germain-en-Laye, France), using the Effectene reagent (QIAGEN, Hilden, Germany), as described previously [25,30]. GC box constructs containing C or the specified modifications at the respective site were transfected in parallel. Cells were split 6 h after transfection and either fixed immediately with 1% of formaldehyde or seeded into separate wells to be fixed after the indicated time intervals, as described previously. Harvested formaldehyde-fixed cells were analyzed by using a FACSCalibur™ flow cytometer and the CellQuest™ Pro software (Beckton Dickinson GmbH, Heidelberg, Germany). The EGFP expression was quantified as median FL1-H fluorescence over the population of transfected cells, defined by the DsRed expression, as described previously [41]. Relative expression levels were calculated for each modification type in the individual experiments based on the expression of the control construct harboring cytosine.

### 5.4. TDG Gene Knockout in HeLa Cells

HeLa-derived clonal cell lines with deletion of a critical portion of the *TDG* gene were generated by a CRISPR-Cas9-mediated gene-editing procedure, using the pX330-sgCas9-HF1 vector (Addgene, Watertown, MA, USA) according to the supplier’s instructions [50], with minor adjustments. Single guide RNA (sgRNA) sequences targeting human *TDG* locus were designed by using CHOPCHOP online tool and subcloned into the BbsI sites [51]. A pair of sgRNAs targeting the exons 2 (sg11029) and 5 (sg16922) was identified as the most efficient, based on screening of four different sgRNAs. The inserts used for cloning of these sgRNAs were obtained by pairwise annealing of synthetic oligonucleotides (Eurofins MWG Operon, Ebersberg, Germany): 5′-CACCGACGAAATATGGACGTTCAAG (sg110292, forward) with 5′-AAACCTTGAACGTCCATATTTCGTC (sg11029, reverse) and 5′-CACCGCTACCAGGGAAGTATGGTAT (sg16922, forward) with 5′-CACCGCTACCAGGGAAGTATGGTAT (sg16922, reverse). Exponentially growing HeLa cells were co-transfected in 6-well plates with the combination of both sgRNA/Cas9 expression vectors (300 ng each) and 50 ng pZAJ vector [52] as a transfection marker. Transfected cells were sorted after 40 h into two 96-well plates, based on the top 5 percentile of the EGFP expression. Single cell sorting was performed in the Flow Cytometry Core Facility of the Institute for Molecular Biology gGmbH (IMB Mainz) under supervision of scientific staff. After two weeks, growing clones were transferred to 25 cm^2^ flasks and screened by PCR for the presence of a deletion spanning the sequence between the sgRNA-targeted sites in the exons 2 and 5 (including the catalytic R140 codon). The primers were 5′-TCCTCTGTAATCCACTCTAA (forward) and 5′-AGCTCAGCTTGAACTAGATA (reverse). Preselected clones were next screened for the presence of non-rearranged *TDG* alleles to eliminate the positives. The primer pair detecting the non-rearranged exon 2 was 5′-TCCTCTGTAATCCACTCTAA (forward) and 5′-ATGTCCCTACTCTGATCTTT (reverse). The remaining clones were expanded and the TDG knockout was validated by Western blotting of protein extracts with a 1:5000 dilution of the TDG rabbit polyclonal antibody #PA5-29140 (Thermo Fisher Scientific Inc.) and a 1:10,000 dilution of the IRDye^®^ 800CW donkey anti-rabbit IgG #926-32213 (LI-COR Biosciences GmbH, Bad Homburg, Germany). Blots were analyzed by using the Odyssey 9120 infrared imaging system (LI-COR). Stripped membranes were subsequently probed with a 1:10,000 dilution of the mouse monoclonal antibody AC88 to HSP90 # ADI-SPA-830 (Enzo Life Sciences GmbH, Lörrach, Germany) and a 1:10,000 dilution of the IRDye^®^ 800CW donkey anti-mouse secondary antibody # 926-68072 (LI-COR).

## Figures and Tables

**Figure 1 ijms-22-11025-f001:**
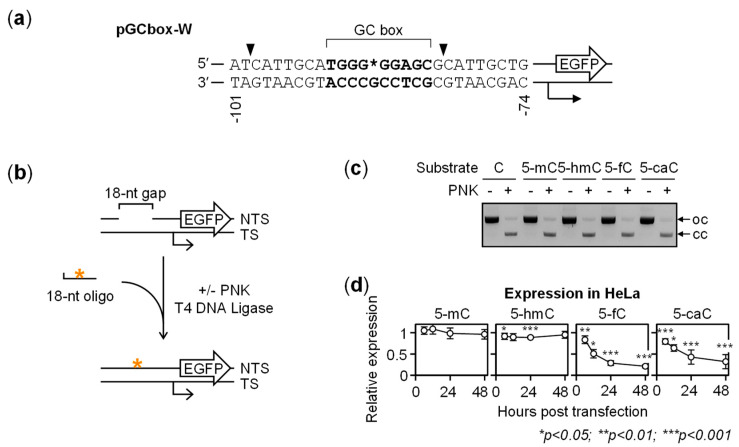
Effects of 5-methylcytosine (5-mC), 5-hydroxymethylcytosine (5-hmC), 5-formylcytosine (5-fC) and 5-carboxycytosine (5-caC) in the purine-rich strand on the GC box activity. (**a**) Scheme of the reporter enhanced green fluorescence protein (*EGFP*) gene under the control of a GC box as the only upstream activating element. Synthetic oligonucleotide containing cytosine modifications at the unique CpG site (position indicated with an asterisk) were incorporated into the gap generated with the Nb.BsrDI nicking endonuclease (cleavage sites indicated with arrowheads). (**b**) Procedure for the incorporation of synthetic oligonucleotides containing C/5-mC/5-hmC/5-fC/5-caC (*) into the gap generated by depletion of the purine-rich strand of the targeted GC box. Aliquots of the same annealing reactions were incubated with or without T4 polynucleotide kinase (PNK) to validate full replacement of the native DNA strand. DNA strand labeling denotes transcribed strand (TS) of the *EGFP* gene and the non-transcribed strand (NTS); broken arrow indicates the transcription start site and direction. (**c**) Agarose gel electrophoresis of the reporter constructs generated by targeted incorporation of C/5-mC/5-hmC/5-fC/5-caC into plasmid DNA. Arrows indicate the open circular (oc) and covalently closed (cc) forms. (**d**) Expression time course of constructs containing 5-mC/5-hmC/5-fC/5-caC in transfected HeLa cells. All values (mean ± SD) are calculated relative to the expression of the control construct harboring synthetic oligonucleotide containing cytosine for *n* = 6 independent experiments (the 12-h point was skipped in three of the experiments). Representative flow cytometry data is shown in Appendix A.

**Figure 2 ijms-22-11025-f002:**
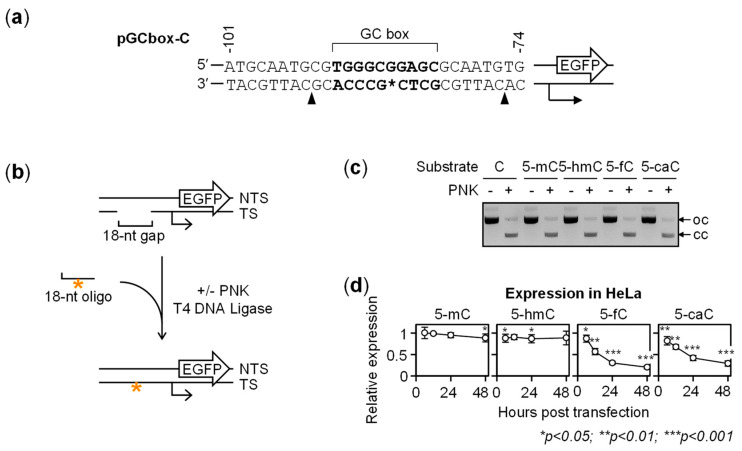
Effects of 5-mC, 5-hmC, 5-fC and 5-caC in the pyrimidine-rich strand of the GC box. (**a**) Promoter sequence with the modified cytosine position (asterisk) and Nb.BsrDI nicking sites (arrowheads). (**b**) Scheme of the incorporation procedure of C/5-mC/5-hmC/5-fC/5-caC (*) into the pyrimidine-rich strand of the GC box. (**c**) Agarose gel electrophoresis of the reporter constructs containing C/5-mC/5-hmC/5-fC/5-caC. The open circular (oc) and covalently closed (cc) forms are indicated by arrows. (**d**) The expression time course in transfected HeLa cells (mean ± SD) for *n* = 6 independent experiments (the 12-h point was skipped in three of them). Representative flow cytometry data are shown in Appendix A.

**Figure 3 ijms-22-11025-f003:**
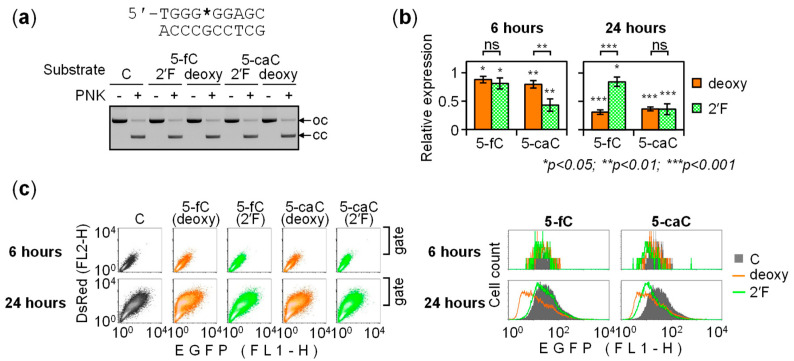
Effects of BER-resistant analogs of 5-fC and 5-caC in the purine-rich strand on the GC box activity. (**a**) Efficient incorporation of 5-fC and 5-caC deoxyribonuceotides (deoxy) and their 2′-(*R*)-fluorinated analogs (2′F) into the purine-rich strand of the GC box. Agarose gel electrophoresis of the fully ligated constructs and the respective “no PNK” controls. The open circular (oc) and covalently closed (cc) forms are indicated by arrows. (**b**) Quantification of the EGFP expression driven by GC box containing 5-fC and 5-caC with or without 2′-fluorination, relative to C, 6 and 24 h post-transfection (mean ± SD, *n* = 4 independent experiments). (**c**) A representative flow cytometry experiment. Cells were gated based on the expression of the transfection marker DsRed, as shown on two-dimensional fluorescence scatter-plots (left panels), and the resulting EGFP signal distributions (right panels, samples overlaid) analyzed to determine the median EGFP fluorescence.

**Figure 4 ijms-22-11025-f004:**
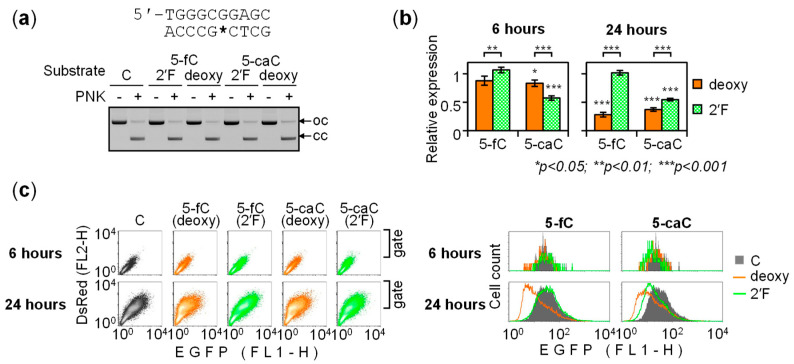
Effects of BER-resistant analogs of 5-fC and 5-caC in the pyrimidine-rich strand on the GC box activity. (**a**) Agarose gel electrophoresis of the fully ligated deoxy and 2′F constructs and the respective “no PNK” controls. The open circular (oc) and covalently closed (cc) forms of the vector are indicated by arrows. (**b**) Relative expression of the 5-fC and 5-caC (deoxy versus 2′F) constructs 6 and 24 h post-transfection (mean ± SD, *n* = 4 independent experiments). (**c**) A representative flow cytometry experiment: two-dimensional fluorescence scatter-plots (left) and the derived EGFP fluorescent distribution plots for the indicated transfection condition (right panels, samples overlaid).

**Figure 5 ijms-22-11025-f005:**
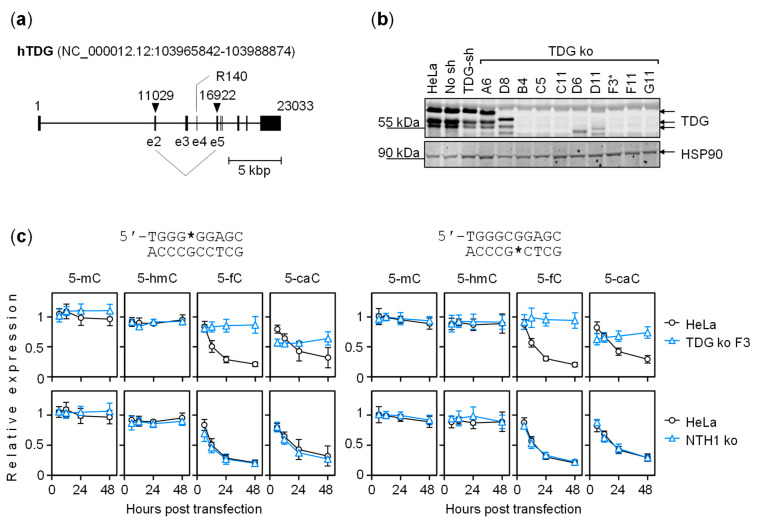
Effect of TDG knockout on the activity of GC box containing 5-mC/5-hmC/5-fC/5-caC. (**a**) Strategy for TDG knockout in Hela cells. The *TDG* gene was targeted simultaneously at two different sites by a pair of single guide RNAs enclosing the codon for the catalytic R140 residue. The Cas9 cut sites are indicated by arrowheads. (**b**) Validation of TDG negative clones by Western blotting. Clone F3 further used as a transfection host for the gene expression analyses is marked (*). (**c**) Expression time course of constructs containing 5-mC/5-hmC/5-fC/5-caC in the purine-rich (left group of plots) or the pyrimidine-rich GC box strand (right). Isogenic clonal TDG knockout (upper row) and NTH1 knockout (lower row) cell lines were compared with the parental HeLa cell line (overlaid in plots). All cell lines were transfected in parallel with the same sets of reporter constructs. Results of *n* = 3 independent experiments (mean ± SD).

**Figure 6 ijms-22-11025-f006:**
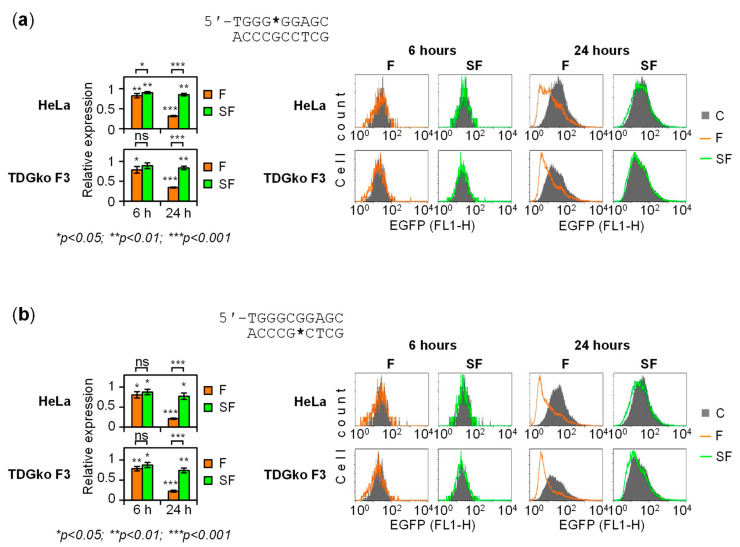
Effect of acytosinic sites on the GC box activity. (**a**) Expression analyses of constructs containing a synthetic abasic site in the purine-rich strand of the GC box. Relative EGFP expression (mean ± SD) for *n* = 4 independent experiments (left) and the representative fluorescent distribution plots (right). Acytosinic modifications used were tetrahydrofuran (F) or a tetrahydrofuran with nuclease resistant phosphorothioate 5′-linkage (SF). (**b**) Analogous analyses of constructs containing F or SF acytosinic sites in the pyrimidine-rich strand.

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
