# Peer review of "Direct and Base Excision Repair-Mediated Regulation of a GC-Rich cis-Element in Response to 5-Formylcytosine and 5-Carboxycytosine"

_ijms, 2021, doi:10.3390/ijms222011025_

Round 1
Reviewer 1 Report
10th September, 2021
Review of the Manuscript ID: ijms-1391943, by N. Müller et al., entitled: “Direct and base excision repair-mediated regulation of a GC-rich cis-element in response to 5-formylcytosine and 5-carboxycytosine” that is intended to be published as the Article in International Journal of Molecular Sciences
(separate Microsoft Word file as Reviewer Attachment for Manuscript ID ijms-1391943 IJMS 10th September 2021 that includes Comments to the Authors is also uploaded)
Taking into consideration research highlight, contribution of the Authors to the progress in the research domain, thorough manner of data presentation, very well writing in English, abundance of Materials and Methods, Results, Figures and Supplementary files (diligent graphic visualization), the quality of this paper deserves praise and merits my support. The Authors have received the high scores from me for the originality, importance of the work and the scientific value of their paper. In my opinion, the current paper provides insightfully interpreting topical and coming trends in the explanation of molecular mechanisms underlying and identification of intrinsic epigenetic factors affecting de novo initiating the expression of transcriptionally suppressed genes. For all those reasons, I strongly recommend the Editorial Board to allow for publication of this very interesting, valuable and comprehensively prepared paper in International Journal of Molecular Sciences, after the minor revision of the manuscript will have been completed by the Authors and provided that the Authors are ready to consider all the Reviewer comments indicated below:
1) There is a lack of the separate Conclusions and Abbreviations sections in the paper. That is why, these sections should have been added by the Authors to the manuscript.
2) The References section has to be prepared in the format compatible with the requirements of International Journal of Molecular Sciences.
General Comment of the Reviewer:
Before the manuscript will have been accepted for publication in International Journal of Molecular Sciences, it requires the minor revision (according to all the remarks and suggestions of the Reviewer).

Author Response
We thank the reviewer for high evaluation of our work expressed in her/his commentary. Below are answers to the specific points.
1) „ There is a lack of the separate Conclusions and Abbreviations sections in the paper. That is why, these sections should have been added by the Authors to the manuscript.“
Response 1: We have outlined the last paragraph of Discussion as a separate Conclusions section (lines 379-396 in the revised version).
We have now defined all abbreviations, following the manuscript preparation instructions: “Acronyms/Abbreviations/Initialisms should be defined the first time they appear in each of three sections: the abstract; the main text; the first figure or table.” to meet this request. Accordingly, we have not added a separate Abbreviations section.
2) “The References section has to be prepared in the format compatible with the requirements of International Journal of Molecular Sciences.”
Response 2: We have followed the references format, as highlighted in the MDPI Reference List and Citations Style Guide. We apologise if any details were missed.
Reviewer 2 Report
Müller et al. in the manuscript “Direct and base excision repair-mediated regulation of a GC-rich cis-element in response to 5-formylcytosine and 5-carboxcytosine” suggest that 5-caC play an inhibiting role in the activation of promoters, underlying the DNA methylation process.
The paper is adequately structured and try to find a role of the 5-mC oxidation products in a GC box cis-element (5′-TGGGCGGAGC) on the gene expression and new insights are necessary in this direction. In my opinion, the paper deserves publication if authors take into account the following observations:
- Paragraph 2.3 Figure 3b-4b Is it possible to quantify the amount of 5-caC at 6 and 24 hours in the transfected plasmid DNA? If not, comment in discussion.
- According to the interpretation of the authors the long term repression of EGFP expression is due to APE1 activity (line 350-352 discussion). Is it possible to measure the amount of nicked plasmid DNA at 6-24 hours? If not, comment this point in discussion.
- Why nicked DNA by APE1 is not rapidly repaired? Indeed 24 hours are not few hours (line 224-226). Clarify this point in discussion.
- Paragraphs 2.4 and 2.5 are not always smooth and should be simplified to make them less difficult for the reader.
- Line 84: eliminate “.”
- Please introduce in fig 1-b legend and in other parts of the text the sense of acronyms (NTS, TS, etc..). A table containing a list of acronyms and meaning should be introduced.
- Line 283: is “purine strand” instead of “pyrimidine strand” correct?
- Parental HeLa cell line instead of paternal HeLa cell line. Please correct line 246, 249, 262, 280
- Please introduce information about Hela cells (code and biosource center) and NTH1 knockout cell line in materials and methods (this last one has never mentioned in the entire manuscript, is human cell line, engineered in authors’ labs?)
Author Response
Response to Reviewer 2 Comments
1) Paragraph 2.3 Figure 3b-4b Is it possible to quantify the amount of 5-caC at 6 and 24 hours in the transfected plasmid DNA? If not, comment in discussion.
Response 1: This is a valid and important point, which we could not clarify in our experimental system. The major problem is that a great part of transfected DNA apparently remains bound to the transfection reagent or is distributed within cells to compartments, where it is not available to repair or transcription. We think, it is for this reason that our attempts to efficiently address repair of 5-caC (or any other modification in transfected plasmid DNA) were not successful. To succeed, it would be necessary to separate or enrich the relevant fraction of repair- and transcription-competent DNA, which we could not achieve so far. As suggested, we now discuss this point in a separate paragraph (lines 365-378 in the revised version).
2) According to the interpretation of the authors the long term repression of EGFP expression is due to APE1 activity (line 350-352 discussion). Is it possible to measure the amount of nicked plasmid DNA at 6-24 hours? If not, comment this point in discussion.
Response 2: We have tried this type of analyses and have got no evidence for persistent strand break at the original modification site. However, quantitative analysis of the completion of BER or detection of the abasic or strand-cleaved intermediates in vector DNA recovered from cells is insensitive for the same reasons, as discussed for 5-caC above (added fragment in Discussion, lines 365-378).
3) Why nicked DNA by APE1 is not rapidly repaired? Indeed 24 hours are not few hours (line 224-226). Clarify this point in discussion.
Response 3: We believe that nicked DNA is, indeed, rapidly repaired. Our current model is that APE1-generated nick would induce silencing, e.g., via an altered chromatin structure, as shown for other BER substrates previously (references *** and ***). Such silencing apparently persists for some time after the completion of repair. However, we have so far no evidence for the pathway connecting the APE1 step with subsequent chromatin rearrangements. This question is hard to approach specifically, because undamaged DNA also undergoes chromatin reorganisation and silencing during the first 20-40 hours after transfection, depending on the host cell line. We hope that we have satisfactorily explained the proposed mechanism, in added discussion (lines 358-360 and 365-378]. A lengthier discussion of indirect evidence from our unpublished results would not be helpful in our opinion. We apologise that we cannot provide a better mechanistic insight at this point.
4) Paragraphs 2.4 and 2.5 are not always smooth and should be simplified to make them less difficult for the reader.”
Response 4: We revised the fragments towards clarity by using shorter sentences in a slightly re-arranged order (lines 257-268, 303-306, 311-312)
5) Line 84: eliminate “.
Response 5: The issue with subtitle 2.1. formatting is corrected now
6) Please introduce in fig 1-b legend and in other parts of the text the sense of acronyms (NTS, TS, etc..). A table containing a list of acronyms and meaning should be introduced.
Response 6: We have defined all acronyms now, as requested. Following the manuscript preparation instructions, we scanned the text to ensure that all acronyms and abbreviations are explained “first time they appear in each of three sections: the abstract; the main text; the first figure or table.” Accordingly, we have not added a separate table of acronyms.
7) Line 283: is “purine strand” instead of “pyrimidine strand” correct?
Response 7: Thank you for noticing the mistake. It is now corrected to “pyrimidine-rich” (line 289).
8) Parental HeLa cell line instead of paternal HeLa cell line. Please correct line 246, 249, 262, 280
Response 8: Thank you for noticing the mistake. All corrected now, as suggested
9) Please introduce information about Hela cells (code and biosource center) and NTH1 knockout cell line in materials and methods (this last one has never mentioned in the entire manuscript, is human cell line, engineered in authors’ labs?)
Response 9: We now included all the requested information (lines 437-440). The NTH1 knockout cell line (yet unpublished) is mentioned in Figure 5 and in the respective paragraph in Results section 2.4. (line 271). As TDG knockout, it was generated in our lab by CRISPR-Cas9 editing of HeLa.